# A Unified Spectral Sparsification Framework for Directed Graphs

## Abstract

Recent *spectral graph sparsification* research allows constructing nearly-linear-sized subgraphs that can well preserve the spectral (structural) properties of the original graph, such as the first few eigenvalues and eigenvectors of the graph Laplacian, leading to the development of a variety of *nearly-linear time* numerical and graph algorithms. However, there is not a unified approach that allows for truly-scalable spectral sparsification of both directed and undirected graphs. For the first time, we prove the existence of linear-sized spectral sparsifiers for general directed graphs, and introduce a practically-efficient yet unified spectral graph sparsification approach that allows sparsifying real-world, large-scale directed and undirected graphs with guaranteed preservation of the original graph spectra. By exploiting a highly-scalable (nearly-linear complexity) spectral matrix perturbation analysis framework for constructing nearly-linear sized (directed) subgraphs, it enables us to well preserve the key eigenvalues and eigenvectors of the original (directed) graph Laplacians. The proposed method has been validated using various kinds of directed graphs obtained from public domain sparse matrix collections, showing promising results for solving directed graph Laplacians, spectral embedding, and partitioning of general directed graphs, as well as approximately computing (personalized) PageRank vectors.

## 1 Introduction

Many research problems for simplifying large graphs leveraging spectral graph theory have been extensively studied by mathematics and theoretical computer science (TCS) researchers in the past decade (Batson et al., 2012; Spielman & Teng, 2011; Kolev & Mehlhorn, 2015; Peng et al., 2015; Lee & Sun, 2017; Cohen et al., 2017; 2018). Recent *spectral graph sparsification* research allows constructing nearly-linear-sized subgraphs that can well preserve the spectral (structural) properties of the original graph, such as the the first few eigenvalues and eigenvectors of the graph Laplacian. The related results can potentially lead to the development of a variety of *nearly-linear time* numerical and graph algorithms for solving large sparse matrices and partial differential equations (PDEs), graph-based semi-supervised learning (SSL), computing the stationary distributions of Markov chains and personalized PageRank vectors, spectral graph partitioning and data clustering, max flow and multi-commodity flow of undirected graphs, nearly-linear time circuit simulation and verification algorithms, etc. (Koutis et al., 2010; Spielman & Teng, 2011; Christiano et al., 2011; Spielman & Teng, 2014; Kelner et al., 2014; Cohen et al., 2017; 2018; Feng, 2016; 2018).

However, there is not a unified approach that allows for truly-scalable spectral sparsification of both directed and undirected graphs. For example, the state-of-the-art sampling-based methods for spectral sparsification are only applicable to undirected graphs (Spielman & Srivastava, 2011; Koutis et al., 2010; Spielman & Teng, 2014); the latest algorithmic breakthrough in spectral sparsification of directed graphs (Cohen et al., 2017; 2018) can only handle strongly-connected directed graphs [1], which inevitably limits its applications when confronting real-world graphs, since many directed graphs may not be strongly connected, such as the graphs used in chip design automation (e.g., timing analysis) tasks as well as the graphs used in machine learning and data mining tasks.

---

[1] A strongly connected directed graph is a directed graph in which any node can be reached from any other node along with direction.

Consequently, there is still a pressing need for the development of highly-robust (theoretically-rigorous) and truly-scalable (nearly-linear complexity) algorithms for reducing real-world large-scale (undirected and directed) graphs while preserving key graph spectral (structural) properties. In summary, we make the following contributions:

- We, for the first time, prove the existence of linear-sized spectral sparsifiers for general directed graphs, and introduces a practically-efficient yet unified spectral sparsification approach that allows simplifying real-world, large-scale directed and undirected graphs with guaranteed preservation of the original graph spectra.

- We exploit a highly-scalable (nearly-linear complexity) spectral matrix perturbation analysis framework for constructing ultra-sparse (directed) subgraphs that can well preserve the key eigenvalues and eigenvectors of the original graph Laplacians. Unlike the prior state-of-the-art methods that are only suitable for handling specific types of graphs (e.g., undirected or strongly-connected directed graphs (Spielman & Srivastava, 2011; Cohen et al., 2017)), the proposed approach is more general and thus will allow for truly-scalable spectral sparsification of a much wider range of real-world complex graphs.

- Through extensive experiments on real-world directed graphs, we show how the proposed directed graph spectral sparsification method can be exploited for computing PageRank vectors, directed graph clustering and developing directed graph Laplacian solvers.

The spectrally-sparsified directed graphs constructed by the proposed approach will potentially lead to the development of much faster numerical and graph-related algorithms. For example, spectrally-sparsified social (data) networks allow for more efficient modeling and analysis of large social (data) networks; spectrally-sparsified neural networks allow for more scalable model training and processing in emerging machine learning tasks; spectrally-sparsified web-graphs allow for much faster computations of personalized PageRank vectors; spectrally-sparsified integrated circuit networks will lead to more efficient partitioning, modeling, simulation, optimization and verification of large chip designs, etc.

## 2 RELATED WORKS

**Directed graph symmetrization.** When dealing with the directed graph sparsification, it's natural to apply symmetrization methods for converting asymmetric directed graphs into symmetric undirected graphs, so that we can apply the existing spectral graph theories for directed graphs after symmetrization. In the following, given a directed graph or its corresponding adjacency matrix [2] $A$, we will review the most popular graph symmetrization methods:

- $\mathbf{A} + \mathbf{A}^\top$ **symmetrization** simply ignores the edges' directions, which is the simplest and most efficient way for directed graph symmetrization. However, edge directions may play an important role in directed graphs. As shown in Figure 1, edges $(8, 1)$ and $(4, 5)$ seem to have the equal importance in the symmetrized undirected graph $A + A^\top$. However, in the original directed graph, edge $(8, 1)$ is much more important than edge $(4, 5)$, since removing edge $(8, 1)$ will lead to the loss of more connections in the directed graph. For example, removing edge $(4, 5)$ will only affect the walks from node 4 to any other nodes and walks from any other nodes to node 5. However, if we remove edge $(8, 1)$ in the directed graph, it will not only affect walks from node 8 to any other nodes and walks from any other nodes to node 1, there will be also no access from node 5 ,6, 7 and 8 to any of nodes 1, 2, 3 and 4.

- **Bibliographic symmetrization** (Satuluri & Parthasarathy, 2011) adopts $AA^\top + A^\top A$ as the adjacency matrix after symmetrization to take the in-going and out-going edges into consideration. However, it cannot be scaled to large-scale graphs since it will create much denser undirected graphs after symmetrization. Also, disconnected graphs can be created due to the $\mathbf{AA}^\top + \mathbf{A}^\top\mathbf{A}$ symmetrization, as shown in Figure 1.

- **Random-walk symmetrization** (Chung, 2005) is based on random walks and allows normalized cut to be preserved after symmetrization. This is also the symmetrization approach used in recent work for spectral sparsification of directed graphs (Cohen et al., 2017). However, it only works on strongly-connected aperiodic directed graphs. For

---

[2] The concept the adjacency matrix for the directed graph will be further introduced in Section 3.1

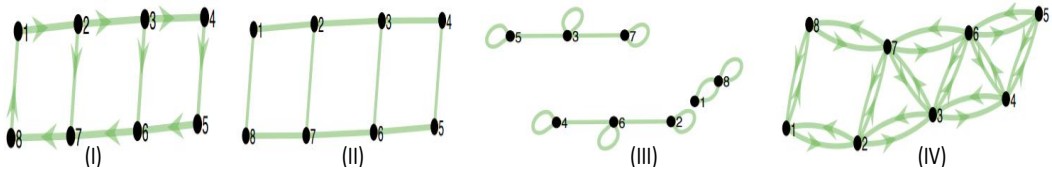

Figure 1: Converting a directed graph $G$ in (I) into undirected graphs using $\mathbf{A} + \mathbf{A}^\top$ as shown in (II), $\mathbf{A}\mathbf{A}^\top + \mathbf{A}^\top\mathbf{A}$ as shown in (III), and the proposed $\mathbf{L_G}\mathbf{L_G}^\top$ symmetrization as shown in (IV).

Table 1: Summary of symbols used in this paper

| Before symmetrization | After symmetrization |
|---|---|
| $G = (V, E_G, w_G)$: directed (undirected) graph | $G_u = (V, E_{G_u}, w_{G_u})$: undirected graph |
| $S = (V, E_S, w_S)$: sparsifier of graph $G$ | $S_u = (V, E_{S_u}, w_{S_u})$: sparsifier of graph $G_u$ |
| $L_G$: Laplacian matrix of graph $G$ | $L_{G_u}$: Laplacian matrix of graph $G_u$ |
| $L_S$: Laplacian matrix of sparsifier $S$ | $L_{S_u}$: Laplacian matrix of sparsifier $S_u$ |

example, we can not apply the random-walk based symmetrization for the directed graph shown in Figure 1, since it's a periodic directed graph.

**Cheeger's inequality for directed graphs.** In undirected graph problems, Cheeger's inequality plays a significant role in spectral analysis of undirected graphs, which connects Cheeger constant (conductance) with spectral properties (eigenvalues of the graph Laplacian matrix) of a graph. In (Chung, 2005) the Cheeger's inequality has been extended to directed graphs based on the random-walk Laplacian symmetrization scheme, as we mentioned earlier. It also provides the bound for the smallest eigenvalue of the directed graph Laplacian. However, the related theoretical results can only be applied to strongly-connected and aperiodic directed graphs, which are rare in real-world applications.

**Spectral sparsification of directed graphs.** The latest algorithmic breakthrough in spectral sparsification for strongly-connected aperiodic graphs has been introduced based on the results in (Chung, 2005), which proposes to convert strongly-connected graphs into Eulerian graphs via Eulerian scaling, and subsequently sparsify the undirected graphs (obtained via directed graph symmetrization (Chung, 2005)) leveraging existing undirected graph spectral sparsification methods (Cohen et al., 2017). It has been shown that such an approach can potentially lead to the development of almost-linear-time algorithms for solving asymmetric linear systems, computing the stationary distribution of a Markov chain and computing expected commute times in a directed graph, etc (Cohen et al., 2017; 2018).

## 3 A THEORETICAL FRAMEWORK FOR UNIFIED SPECTRAL SPARSIFICATION

### 3.1 LAPLACIANS FOR DIRECTED AND UNDIRECTED GRAPHS

Consider a directed graph $G = (V, E_G, w_G)$ with $V$ denoting the set of vertices, $E_G$ representing the set of directed edges, and $w_G$ denoting the associated edge weights. Let $n = |V|$, $m = |E_G|$ be the size of node and edge set. In the following, we denote the diagonal matrix by $\mathbf{D_G}$ with $D_G(i,i)$ being equal to the (weighted) outdegree of node $i$, as well as the adjacency matrix of $G$ by $\mathbf{A_G}$:

$$A_G(i,j) = \begin{cases} w_G(i,j) & \text{if } (i,j) \in E_G \\ 0 & \text{otherwise}. \end{cases} \tag{1}$$

Then the directed Laplacian matrix can be constructed as follows (Cohen et al., 2017): $\mathbf{L_G} = \mathbf{D_G} - \mathbf{A_G}^\top$. For better illustration, we have summarized symbols used in our paper in Table 1.

It can be shown that any directed (undirected) graph Laplacian constructed using (3.1) will satisfy the following properties: **I)** Each column (and row) sum is equal to zero; **II)** All off-diagonal elements are non-positive; **III)** The Laplacian matrix is asymmetric (symmetric) and indefinite (positive semidefinite).

## 3.2 SPECTRAL SPARSIFICATION OF (UN)DIRECTED GRAPHS

Graph sparsification aims to find a subgraph (sparsifier) $S = (V, E_S, w_S)$ that has the same set of vertices but much fewer edges than the original graph $G$. There are two types of sparsification methods: the cut sparsification methods preserve cuts in the original graph through random sampling of edges (Benczúr & Karger, 1996), whereas spectral sparsification methods preserve the graph spectral (structural) properties, such as distances between vertices, effective resistances, cuts in the graph, as well as the stationary distributions of Markov chains (Cohen et al., 2017; 2018; Spielman & Teng, 2011). Therefore, spectral graph sparsification is a much stronger notion than cut sparsification.

**For undirected graphs**, spectral sparsification aims to find an ultra-sparse subgraph proxy that is spectrally-similar to the original one. $G$ and $S$ are said to be $\sigma$-spectrally similar if the following condition holds for all real vectors $\mathbf{x} \in \mathbb{R}^V$:

$$\frac{\mathbf{x}^\top \mathbf{L_S} \mathbf{x}}{\sigma} \leq \mathbf{x}^\top \mathbf{L_G} \mathbf{x} \leq \sigma \mathbf{x}^\top \mathbf{L_S} \mathbf{x}, \tag{2}$$

where $\mathbf{L_G}$ and $\mathbf{L_S}$ denote the symmetric diagonally dominant (SDD) Laplacian matrices of graphs $G$ and $S$, respectively. Relative condition number can be defined as $\kappa(\mathbf{L_G}, \mathbf{L_S}) \leq \sigma^2$, implying that a smaller relative condition number or $\sigma^2$ corresponds to a higher (better) spectral similarity between two graphs.

**For directed graphs : Spectrum-preserving $\mathbf{L_G} \mathbf{L_G}^\top$ symmetrization** the subgraph $S$ can be considered spectrally similar to the original graph $G$ if the condition number or the ratio between the largest and smallest singular values of $\mathbf{L_S}^+ \mathbf{L_G}$ is close to 1 (Cohen et al., 2017; 2018), where $\mathbf{L_S}^+$ denotes the Moore-Penrose pseudoinverse of $\mathbf{L_S}$. Spectral sparsification of directed graphs is equivalent to finding an ultra-sparse subgraph $S$ such that the condition number of $(\mathbf{L_S}^+ \mathbf{L_G})^\top (\mathbf{L_S}^+ \mathbf{L_G})$ is small enough. Since the singular values of $\mathbf{L_S}^+ \mathbf{L_G}$ are the square roots of eigenvalues of $(\mathbf{L_S}^+ \mathbf{L_G})^\top (\mathbf{L_S}^+ \mathbf{L_G})$.

While $(\mathbf{L_S}^+ \mathbf{L_G})^\top (\mathbf{L_S}^+ \mathbf{L_G})$ can be written into $\mathbf{L_G}^\top (\mathbf{L_S} \mathbf{L_S}^\top)^+ \mathbf{L_G}$, $\mathbf{L_G}^\top (\mathbf{L_S} \mathbf{L_S}^\top)^+ \mathbf{L_G}$ is not equal to $(\mathbf{L_S} \mathbf{L_S}^\top)^+ (\mathbf{L_G} \mathbf{L_G}^\top)$. They do share the same eigenvalues under special conditions according to the following theorem (Horn & Johnson, 2012):

**Theorem 3.1.** *Suppose that matrices $\mathbf{X} \in \mathbb{R}^{m',n'}$ and $\mathbf{Y} \in \mathbb{R}^{n',m'}$ with $m' \leq n'$. Then the $n'$ eigenvalues of $\mathbf{YX}$ are the $m'$ eigenvalues of $\mathbf{XY}$ together with $n' - m'$ zeroes; that is $p_{\mathbf{YX}}(t) = t^{n'-m'} p_{\mathbf{XY}}(t)$. If $m' = n'$ and at least one of $\mathbf{X}$ or $\mathbf{Y}$ is nonsingular, then $\mathbf{XY}$ and $\mathbf{YX}$ are similar.*

Based on Therorem 3.1, $\mathbf{L_G}^\top (\mathbf{L_S} \mathbf{L_S}^\top)^+ \mathbf{L_G}$ and $(\mathbf{L_S} \mathbf{L_S}^\top)^+ (\mathbf{L_G} \mathbf{L_G}^\top)$ share the same eigenvalues when a small value is added on each diagonal of $\mathbf{L_G}$. Under this condition, spectral sparsification of directed graphs is equivalent to finding an ultra-sparse subgraph $S$ such that the condition number of $(\mathbf{L_S} \mathbf{L_S}^\top)^+ (\mathbf{L_G} \mathbf{L_G}^\top)$ is small enough. Theorem 3.2 shows both $\mathbf{L_G} \mathbf{L_G}^\top$ and $\mathbf{L_S} \mathbf{L_S}^\top$ are the Laplacian matrices for some undirected graphs.

**Theorem 3.2.** *For any directed graph $G = (V, E_G, w_G)$ and its directed Laplacian $\mathbf{L_G}$, its symmetrized undirected graph $G_u = (V, E_{G_u}, w_{G_u})$ can be obtained via Laplacian symmetrization $\mathbf{L_{G_u}} = \mathbf{L_G} \mathbf{L_G}^\top$. $\mathbf{L_{G_u}}$ is positive semi-definite (PSD) and will have the all-one vector as its null space, while the corresponding undirected graph may include negative edge weights.*

The proof is in the Appendix. If we can prove that there exists an ultra-sparse subgraph $S$ such that its corresponding undirected graph $S_u$ (with $\mathbf{L_{S_u}} = \mathbf{L_S} \mathbf{L_S}^\top$) is the spectral sparsifier of $G_u$ (with $\mathbf{L_{G_u}} = \mathbf{L_G} \mathbf{L_G}^\top$), then the directed subgraph $S$ becomes the spectral sparsifier of $G$. More detailed proofs are shown in the Appendix. The core idea of our approach is to leverage a novel spectrum-preserving Laplacian symmetrization procedure to convert directed graphs into undirected ones that may have negative-weighted edges (as shown in Figure 1). Such a Laplacian symmetrization scheme will immediately allow us to exploit existing methods for spectral sparsification.

## 4 A PRACTICAL FRAMEWORK FOR UNIFIED SPECTRAL SPARSIFICATION

To apply our theoretical results to deal with real-world directed graphs, the following concerns should be addressed in advance:

- The undirected graph $\mathbf{L_G} \mathbf{L_G}^\top$ may become too dense to compute and thus may impose high cost during spectral sparsification.

- It can be quite challenging to convert the sparsified undirected graph to its corresponding directed sparsifier $\mathbf{L_S}$, even when $\mathbf{L_{S_u}}$ is available.

To address the above concerns for unified spectral graph sparsification, we propose a practically-efficient framework with following desired features: **1)** our approach dose not require to explicitly compute $\mathbf{L_G}\mathbf{L_G^\top}$ but only the matrix-vector multiplications; **2)** our approach can effectively identify the most spectrally-critical edges for dramatically decreasing the relative condition number; **3)** although our approach requires to compute $\mathbf{L_S}\mathbf{L_S^\top}$, the $\mathbf{L_{S_u}}$ matrix density can be effectively controlled by carefully pruning spectrally-similar edges through the proposed edge similarity checking scheme.

## 4.1 INITIAL SUBGRAPH SPARSIFIER CONSTRUCTION

Motivated by the recent research on low-stretch spanning trees (Elkin et al., 2008; Abraham & Neiman, 2012) and spectral perturbation analysis (Feng, 2016; 2018) for nearly-linear-time spectral sparsification of undirected graphs, we propose a practically-efficient algorithm for sparsifying general directed graphs by first constructing the initial subgraph sparsifiers of directed graphs with the following procedure:

- Compute $\mathbf{D^{-1}}(\mathbf{A_G} + \mathbf{A_G^\top})$ as a new adjacency matrix, where $\mathbf{D}$ denotes the diagonal matrix with each element equal to the row (column) sum of $(\mathbf{A_G} + \mathbf{A_G^\top})$. Recent research shows such split transformations can effectively reduce graph irregularity while preserving critical graph connectivity, distance between node pairs, the minimal edge weight in the path, as well as outdegrees and indegrees when using push-based and pull-based vertex-centric programming (Nodehi Sabet et al., 2018).

- Construct a maximum spanning tree (MST) based on $\mathbf{D^{-1}}(\mathbf{A_G} + \mathbf{A_G^\top})$, which allows us to effectively control the number of outgoing edges for each node so that the resultant undirected graph after Laplacian symmetrization will not be too dense.

- Recover the direction of each edge in the MST and make sure each node of its sparsifier has at least one outgoing edge if there are more than one in the original graph for achieving stronger connectivity in the initial directed sparsifier.

## 4.2 SPECTRAL SPARSIFICATION VIA RIEMANNIAN DISTANCE MINIMIZATION

The Riemannian distance $\delta_2$ between positive definite (PSD) matrices is arguably the most natural and useful distance on the positive definite cone $\mathbb{S}_{++}^n$ (Bonnabel & Sepulchre, 2010), which can be computed by (Lim et al., 2019):

$$\delta_2 : S_{++}^n \times S_{++}^n \to R_+ \quad \text{and} \quad \delta_2(\mathbf{L_{S_u}}, \mathbf{L_{G_u}}) = \left[ \sum_{i=1}^n \log^2 \lambda_i \right]^{\frac{1}{2}}, \tag{3}$$

where $\lambda_{\max} = \lambda_1 \geq \cdots \geq \lambda_{n'} \geq 1 \geq \lambda_{n'+1} \geq, \cdots, \geq \lambda_n$ denote the descending eigenvalues of $\mathbf{L_{S_u}^+}\mathbf{L_{G_u}}$, and $\mathbf{v_1}, \mathbf{v_2}, \cdots, \mathbf{v_n}$ denote the corresponding eigenvectors. Since both $S_u$ and $G_u$ are PSD matrices, we have $\lambda_i \geq 0$, which leads to the following inequality:

$$\delta_2(\mathbf{L_{S_u}}, \mathbf{L_{G_u}}) \leq \sum_{i=1}^{n'} \log \lambda_i + \sum_{i=n'+1}^{n} \log \frac{1}{\lambda_i} \leq \max\{n \log \lambda_1, n \log \frac{1}{\lambda_n}\}. \tag{4}$$

The **Courant-Fischer theorem** allows computing generalized eigenvalues and eigenvectors by:

$$\lambda_1 = \max_{\substack{|x| \neq 0, \\ x^\top \mathbf{1} = 0}} \frac{x^\top L_{G_u} x}{x^\top L_{S_u} x} \qquad \lambda_n = \min_{\substack{|x| \neq 0, \\ x^\top \mathbf{1} = 0}} \frac{x^\top L_{G_u} x}{x^\top L_{S_u} x} \tag{5}$$

where $\mathbf{1} \in \mathbb{R}^n$ denotes the all-one vector. Assigning each node in the graph with an integer value either 0 or 1, the corresponding Laplacian quadratic form measures the boundary size (cut) of a node set. For example, if a node set $Q$ is defined as a subset when its nodes are all assigned with an integer value 1 while other nodes are assigned with a value 0, then node set $Q$ and its boundary $\partial_{G_u}(Q)$ can be represented as

$$Q = \{p \in V : x(p) = 1\} \quad \text{and} \quad \partial_{G_u}(Q) = \{(p, q) \in E, p \in Q, q \notin Q\}. \tag{6}$$

Since the number of outgoing edges crossing the boundary can be computed by $x^\top L_{G_u} x = |\partial_{G_u}(Q)|$, the following holds:

$$\lambda_1 = \max_{\substack{|x| \neq 0, \\ x^\top \mathbf{1} = 0}} \frac{x^\top L_{G_u} x}{x^\top L_{S_u} x} \geq \max_{\substack{|x| \neq 0, \\ x(p) \in \{0,1\}}} \frac{x^\top L_{G_u} x}{x^\top L_{S_u} x} = \frac{|\partial_{G_u}(Q)|}{|\partial_{S_u}(Q)|}, \tag{7}$$

which implies that the maximum mismatch between $G_u$ and $S_u$ will be bounded by $\lambda_1$. Leveraging the dominant generalized eigenvectors will allow us to identify the edges crossing the **maximally-mismatched boundary** $\partial_{G_u}(Q)$. Including such crossing edges into $S_u$ will dramatically decrease the maximum mismatch ($\lambda_1$), thereby improving spectral approximation of the sparsifier. Consequently, the following problem formulation for spectral graph sparsification is proposed:

$$\min_{L_{S_u}} \left\{ \max_x \left( \frac{x^\top L_{G_u} x}{x^\top L_{S_u} x} \right) + \beta \|L_{S_u}\|_1 \right\}, \tag{8}$$

which allows effectively minimizing the largest generalized eigenvalue (upper bound of mismatch) and the Riemaninan distance ($\delta_2$) by including the minimum amount of edges into the subgraph $S_u$.

## 4.3 A UNIFIED SPECTRAL PERTURBATION ANALYSIS FRAMEWORK

As aforementioned, spectral sparsification of (un)directed graphs can be effectively achieved by solving (8). To this end, we will exploit the following spectral perturbation analysis framework. Given the generalized eigenvalue problem $\mathbf{L_{G_u} v_i} = \lambda_i \mathbf{L_{S_u} v_i}$ with $i = 1, \cdots, n$, let matrix $\mathbf{V} = [\mathbf{v_1}, \cdots, \mathbf{v_n}]$. Then $\mathbf{v_i}$ and $\lambda_i$ can be computed to satisfy the following orthogonality requirement:

$$\mathbf{v_i}^\top \mathbf{L_{G_u} v_j} = \begin{cases} \lambda_i, & i = j \\ 0, & i \neq j \end{cases} \quad and \quad \mathbf{v_i}^\top \mathbf{L_{S_u} v_j} = \begin{cases} 1, & i = j \\ 0, & i \neq j. \end{cases} \tag{9}$$

Consider the following first-order generalized eigenvalue perturbation problem:

$$\mathbf{L_{G_u}}(\mathbf{v_i} + \delta \mathbf{v_i}) = (\lambda_i + \delta \lambda_i)(\mathbf{L_{S_u}} + \delta \mathbf{L_{S_u}})(\mathbf{v_i} + \delta \mathbf{v_i}), \tag{10}$$

where a small perturbation $\delta \mathbf{L_{S_u}}$ in $\mathbf{L_{S_u}}$ is introduced, leading to the perturbed generalized eigenvalues and eigenvectors $\lambda_i + \delta \lambda_i$ and $\mathbf{v_i} + \delta \mathbf{v_i}$. By only keeping the first-order terms, (10) becomes:

$$\mathbf{L_{G_u}} \delta \mathbf{v_i} = \lambda_i \mathbf{L_{S_u}} \delta \mathbf{v_i} + \lambda_i \delta \mathbf{L_{S_u}} \mathbf{v_i} + \delta \lambda_i \mathbf{L_{S_u}} \mathbf{v_i}. \tag{11}$$

Let $\delta \mathbf{v_i} = \sum_\mathbf{j} \psi_{\mathbf{i,j}} \mathbf{v_j}$, then (11) can be expressed as:

$$\sum_\mathbf{j} \psi_{\mathbf{i,j}} \mathbf{L_{G_u} v_j} = \lambda_i \mathbf{L_{S_u}} (\sum_\mathbf{j} \psi_{\mathbf{i,j}} \mathbf{v_j}) + \lambda_i \delta \mathbf{L_{S_u}} \mathbf{v_i} + \delta \lambda_i \mathbf{L_{S_u}} \mathbf{v_i} \tag{12}$$

Based on the orthogonality properties in (9), multiplying $v_i$ to both sides of (12) results in

$$\lambda_\mathbf{i} \delta \mathbf{L_{S_u}} \mathbf{v_i} + \delta \lambda_\mathbf{i} \mathbf{L_{S_u}} \mathbf{v_i} = \mathbf{0}. \tag{13}$$

Then the task of spectral sparsification of general (un)directed graphs will require to recover as few as possible extra edges to the initial directed subgraph $S$ such that the largest eigenvalues, or the condition number of $\mathbf{L_{S_u}^+ L_{G_u}}$ can be dramatically mitigated. Expanding (13) will simply lead to:

$$\frac{\delta \lambda_i}{\lambda_i} = -\mathbf{v_i}^\top \delta \mathbf{L_{S_u}} \mathbf{v_i}. \tag{14}$$

Expand $\delta \mathbf{L_{S_u}}$ with only the first-order terms as $\delta \mathbf{L_{S_u}} = \delta \mathbf{L_S} \mathbf{L_S^\top} + \mathbf{L_S} \delta \mathbf{L_S^\top}$, where $\delta \mathbf{L_S} = w_G(p,q) \mathbf{e_{p,q}} \mathbf{e_p}^\top$ for $(p,q) \in E_G \setminus E_S$, $\mathbf{e_p} \in \mathbb{R}^n$ denotes the vector with only the $p$-th element being 1 and others being 0, and $\mathbf{e_{p,q}} = \mathbf{e_p} - \mathbf{e_q}$. The **spectral sensitivity** for each off-subgraph edge $(p,q)$ can be expressed as:

$$\zeta_{p,q} = \mathbf{v_i}^\top (\delta \mathbf{L_S} \mathbf{L_S^\top} + \mathbf{L_S} \delta \mathbf{L_S^\top}) \mathbf{v_i}, \tag{15}$$

It is obvious that (15) can be leveraged to rank the spectral importance of each edge. Consequently, spectral sparsification of general graphs can be achieved by only recovering a few dissimilar edges

with large sensitivity values. In this work, the following method based on t-step power iterations is proposed to achieve efficient computation of dominant generalized eigenvectors

$$\mathbf{v_1} \approx \mathbf{h_t} = (\mathbf{L_{S_u}^+} \mathbf{L_{G_u}})^{\mathbf{t}} \mathbf{h_0}, \tag{16}$$

where $\mathbf{h_0}$ denotes a random vector. When the number of power iterations is small (e.g., $t \leq 3$), $\mathbf{h_t}$ will be a linear combination of the first few dominant generalized eigenvectors corresponding to the largest few eigenvalues. Then the spectral sensitivity for the off-subgraph edge $(p, q)$ can be approximately computed by

$$\zeta_{p,q} \approx \mathbf{h_t^\top} (\delta \mathbf{L_S L_S^\top} + \mathbf{L_S} \delta \mathbf{L_S^\top}) \mathbf{h_t}. \tag{17}$$

The computation of $\mathbf{h_t}$ through power iterations requires solving the linear system of equations $\mathbf{L_{S_u}} \mathbf{x} = \mathbf{b}$ for $t$ times. We note that only $\mathbf{L_{S_u}}$ need to be explicitly computed for generalized power iterations. The latest Lean Algebraic Multigrid (LAMG) solver has been leveraged for computing $\mathbf{h_t}$, which can handle undirected graphs with negative edge weights and achieve $O(m)$ runtime complexity for solving large graph Laplacian matrices (Livne & Brandt, 2012).

### 4.4 EDGE SPECTRAL SIMILARITIES

To avoid recovering redundant edges into the subgraph, it is indispensable to check spectral similarities between candidate off-subgraph edges. In other words, only the off-subgraph edges that are spectrally critical (have higher spectral sensitivity scores) but not spectrally-similar to each other will be recovered to the initial subgraph. To this end, we exploit the following spectral embedding of off-subgraph edges using approximate dominant generalized eigenvectors $\mathbf{h_t}$ computed by (16):

$$\tau_{p,q} = \mathbf{h_t^\top} \mathbf{e_{pq}} \mathbf{e_p^\top} \mathbf{L_S^\top} \mathbf{h_t}, \tag{18}$$

which will help estimate spectral similarities among different off-subgraph edges. To improve the accuracy, we can always compute $r = O(\log n)$ approximate dominant generalized eigenvectors $\mathbf{h_t^{(1)}}, ..., \mathbf{h_t^{(r)}}$ to obtain a $r$-dimensional spectral embedding vector $\mathrm{T}_{p,q}$ for each edge $(p, q)$. The edge spectral similarity between two edges $(p_i, q_i)$ and $(p_j, q_j)$ is defined as follows:

$$\beta_{i,j} = ||\mathrm{T_{p_i,q_i}} - \mathrm{T_{p_j,q_j}}|| / \max(||\mathrm{T_{p_i,q_i}}||, ||\mathrm{T_{p_j,q_j}}||). \tag{19}$$

If $(1 - \beta_{i,j}) < \varrho$ for a given constant $\varrho$, edge $(p_i, q_i)$ is considered spectrally dissimilar with edge $(p_j, q_j)$.

### 4.5 ALGORITHM FLOW AND COMPLEXITY

---

**Algorithm 1** Algorithm Flow for Directed Graph Sparsification

**Input:** $\mathbf{L_G}$, $\mathbf{L_S}$, $d_{\text{out}}$, $\text{iter}_{\text{max}}$, $\lambda_{\text{limit}}$, $\alpha$, $\varrho$

1: Calculate largest generalized eigenvector $\mathbf{h_t}$, largest generalized eigenvalue $\lambda_{\text{max}}$ and let iter $= 1$;
2: **while** $\lambda_{\text{max}} < \lambda_{\text{limit}}$, iter $<$ iter$_{\text{max}}$ **do**
3:     Calculate the spectral sensitivities $\zeta_{p,q}$ for each off-subgraph edges $(p, q) \in E_{G \backslash S}$;
4:     Sort spectral sensitivities in descending order and obtain the top $\alpha\%$ off-subgraph edges into edge list $E_{\text{list}} = [(p_1, q_1), (p_2, q_2), ...]$;
5:     Do $E_{\text{addlist}} = \text{Edge\_Similarities\_Checking}(E_{\text{list}}, \mathbf{L_G}, \mathbf{L_S}, d_{\text{out}}, \varrho)$ ;
6:     Update $S_{\text{new}} = S + E_{\text{addlist}}$ and calculate largest generalized eigenvector $\mathbf{h_{tnew}}$, largest generalized eigenvalue $\lambda_{\text{maxnew}}$ based on $\mathbf{L_G}$ and $\mathbf{L_{Snew}}$ ;
7:     **if** $\lambda_{\text{maxnew}} < \lambda_{\text{max}}$ **then**
8:         Update $S = S_{\text{new}}$, $\mathbf{h_t} = \mathbf{h_{tnew}}$, $\lambda_{\text{max}} = \lambda_{\text{maxnew}}$ ;
9:     **end if**
10:    iter $=$ iter $+ 1$;
11: **end while**
12: Return graph $S$ and $\mathbf{L}_S$.

---

Algorithm 1 shows the algorithm flow for directed graph sparsification, where $\mathbf{L_G}$ is the Laplacian matrix for original graph, $\mathbf{L_S}$ is the Laplacian matrix of initial spanning tree, $d_{\text{out}}$ is the user-defined outgoing degree for nodes, iter$_{\text{max}}$ is the maximum number of iterations, and $\lambda_{\text{limit}}$ is the desired maximum generalized eigenvalue. $E_{\text{addlist}} = \text{Edge\_Similarities\_Checking}(E_{\text{list}}, \mathbf{L_G}, \mathbf{L_S}, d_{\text{out}}, \varrho)$ is introduced in the Appendix. The complexity has been summarized as follows:

Table 2: Results of directed graph spectral sparsification

| Test Cases | $|V_G|$ | $|E_G|$ | $\frac{|E_{S^0}|}{|E_G|}$ | $\frac{|E_S|}{|E_G|}$ | time (s) | $\frac{\lambda_1}{\lambda_{1,fin}}$ |
|---|---|---|---|---|---|---|
| gre_115 | 1.1E2 | 4.2E2 | 0.46 | 0.71 | 0.05 | 8.2E4X |
| gre_185 | 1.8E2 | 1.0E3 | 0.33 | 0.46 | 0.14 | 9.8E3X |
| harvard500 | 0.5E3 | 2.6E3 | 0.31 | 0.40 | 0.64 | 1.2E3X |
| cell1 | 0.7E4 | 3.0E4 | 0.31 | 0.57 | 3.10 | 1.0E5X |
| pesa | 1.2E4 | 8.0E4 | 0.27 | 0.51 | 8.80 | 5.3E8X |
| big | 1.3E4 | 0.9E5 | 0.27 | 0.49 | 12.86 | 4.1E11X |
| gre_1107 | 1.1E3 | 5.6E3 | 0.26 | 0.39 | 0.24 | 58X |
| wordnet3 | 0.8E5 | 1.3E5 | 0.64 | 0.84 | 50.00 | 12X |
| p2p-Gnutella31 | 0.6E5 | 1.5E5 | 0.35 | 0.43 | 11.90 | 6X |
| p2p-Gnutella05 | 8.8E3 | 3.2E4 | 0.23 | 0.65 | 27.59 | 35X |
| mathworks100 | 1.0E2 | 5.5E2 | 0.20 | 0.50 | 0.04 | 30X |

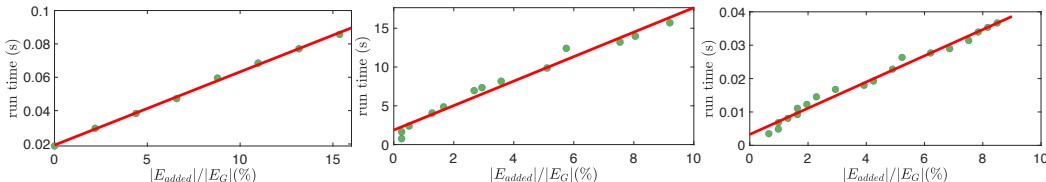

Figure 2: Runtime scalability for gre_1107 (left), big (middle), gre_115 (right)

(a) Generate an initial subgraph $S$ from the original directed graph in $O(m \log n)$ or $O(m + n \log n)$ time;

(b) Compute the approximate dominant eigenvector $\mathbf{h_t}$ and the spectral sensitivity of each off-subgraph edge in $O(m)$ time;

(c) Recover a small amount of spectrally-dissimilar off-subgraph edges into the latest subgraph $S$ according to their spectral sensitivities and similarities in $O(m)$ time;

(d) Repeat steps **(b)** and **(c)** until the desired condition number or spectral similarity is achieved.

## 5 EXPERIMENTAL RESULTS

The proposed algorithm for spectral sparsification of directed graphs has been implemented using MATLAB and C++. Extensive experiments have been conducted to evaluate the proposed method with various types of directed graphs obtained from public-domain data sets (Davis & Hu, 2011).

Table 2 shows comprehensive results on directed graph spectral sparsification for a variety of real-world directed graphs using the proposed method, where $|V_G|(|E_G|)$ denotes the number of nodes (edges) for the original directed graph $G$; $|E_{S^0}|$ and $|E_S|$ denote the numbers of edges in the initial subgraph $S^0$ and final spectral sparsifier $S$. Notice that we will directly apply the Matlab's "eigs" function if the size of the graph is relatively small ($|E_{S^0}| < 1E4$); otherwise we will switch to the LAMG solver for better efficiency when calculating the approximate generalized eigenvector $\mathbf{h_t}$. We report the total runtime for the eigsolver using either the LAMG solver or "eigs" function. $\frac{\lambda_1}{\lambda_{1,fin}}$ denotes the reduction rate of the largest generalized eigenvalue of $\mathbf{L_{S_u}^+ L_{G_u}}$.

Figure 2 shows the runtime scalability regarding to the number of off-subgraph edges ($|E_{added}|$) added in the final sparsifier for graph gre_1107 (left), big (middle) and gre_115 (right). As we can see, the runtime scales linearly with the added number of edges for all three graphs.

Since there are no other existing directed graph sparsification methods to be compared, we compare our proposed method with the existing undirected graph sparsification solver GRASS (Feng, 2016; 2018). To make sure the directed graphs can be applied to the GRASS, we first convert the directed graphs into undirected ones ($G'_u$) using $A + A^\top$ symmetrization. Then undirected graph sparsifiers $S'_u$ will be generated by GRASS. Finally, the final directed graph sparsifiers can be formed by adding the edge directions to the obtained undirected sparsifiers $S'_u$. The experimental results are shown in Table 3, where $\kappa$ represents the relative condition number between the original graph and its final

Table 3: Comparison of spectral sparsification results between the proposed method and GRASS.

| Test cases | GRASS | | | | Our method | |
|---|---|---|---|---|---|---|
| | $\frac{\|E_{S'_u}\|}{\|E_{G'_u}\|}$ | $\kappa(\mathbf{L_{G'_u}}, \mathbf{L_{S'_u}})$ | $\frac{\|E_S\|}{\|E_G\|}$ | $\kappa(\mathbf{L_G}, \mathbf{L_S})$ | $\frac{\|E_S\|}{\|E_G\|}$ | $\kappa(\mathbf{L_G}, \mathbf{L_S})$ |
| gre_115 | 0.92 | 28 | 0.44 | 3760 | 0.43 | **522** |
| gre_185 | 0.67 | 25 | 0.40 | 1140 | 0.41 | **170** |
| gre_1107 | 0.86 | 9 | 0.43 | 2790 | 0.43 | **147** |
| harvard500 | 0.36 | 13 | 0.39 | 5.22E5 | 0.36 | **2.40E4** |
| p2p-Gnutella05 | 0.56 | 6 | 0.57 | 3.90E5 | 0.54 | **1.90E5** |
| p2p-Gnutella31 | 0.68 | 3 | 0.68 | 1.0E5 | 0.68 | **8.20E4** |
| big | 0.60 | 7 | 0.60 | 8803 | 0.60 | **270** |
| wordnet3 | 0.78 | 8 | 0.79 | 3.06E4 | 0.78 | **2.74E3** |

Table 4: Comparison of GMRES results.

| Test cases | $relres$ | No preconditioner | ILU preconditioner | | Spar. preconditioner | | |
|---|---|---|---|---|---|---|---|
| | | $iter$ | $iter$ | $nnz$ | $iter$ | $nnz$ | $\kappa(L_G, L_S)$ |
| gre_115 | 1E-7 | 64 | 22 | 536 | **17** | **336** | 16 |
| gre_185 | 1E-7 | 53 | 27 | 1,190 | **23** | **768** | 19 |
| gre_1107 | 1E-7 | 118 | 24 | 6,771 | **22** | **4,327** | 18 |
| harvard500 | 1E-7 | 80 | 25 | 3,563 | **18** | **2,527** | 20 |
| big | 1E-7 | 356 | 163 | 104,674 | **64** | **72,509** | 108 |
| peta | 1E-7 | $NC$ | 241 | 91,304 | **85** | **67,007** | 28 |

sparsifier. By keeping the similar number of edges in the sparsifiers, we can observe that our method can achieve much better sparsifiers than GRASS does.

More results regarding the directed graph sparsification are shown in Appendix.

The directed graph sparsifier can also be directly utilized as the preconditioner for directed Laplacian solver when solving the linear system equation $\mathbf{Lx} = \mathbf{b}$ with iterative solvers such as generalized minimal residual method (GMRES) (Saad & Schultz, 1986). Table 4 shows comprehensive results for GMRES iterations when no preconditioner is applied, Incomplete LU factorization (ILU) as the preconditioner is applied, and the directed sparsifier Laplacian $\mathbf{L_S}$ as the preconditioner is applied. The MATLAB functions gmres and ilu with default setting are applied in our experiments. "$relres$" is the relative residual to be achieve for three methods, "$iter$" is GMRES iteration number, "$nnz$" is the number of non-zeros in the preconditioner, and "$NC$" represents it did not converge when reaching maximum number of iterations (which is 500 in the experiments). We can conclude that GMRES with directed graph sparsifier as the preconditioner has better convergence rate when comparing to the other two methods. Meanwhile, the sparsifier has much fewer number of non-zeros than ILU preconditioner. More results on directed graph Laplacian solver can be found in Appendix.

In the end, we also demonstrate the applications of our proposed directed graph sparsification in solving directed graph Laplacians using both direct method and iterative method, as well as the applications in computing (Personalized) PageRank vectors and directed graph partitioning in Appendix.

## 6 CONCLUSIONS

For the first time, this paper proves the existence of linear-sized spectral sparsifiers for general directed graphs, and proposes a practically-efficient yet unified spectral graph sparsification framework. Such a novel spectral sparsification approach allows sparsifying real-world, large-scale directed and undirected graphs with guaranteed preservation of the original graph spectral properties. By exploiting a highly-scalable (nearly-linear complexity) spectral matrix perturbation analysis framework for constructing nearly-linear sized (directed) subgraphs, it enables us to well preserve the key eigenvalues and eigenvectors of the original (directed) graph Laplacians. The proposed method has been validated using various kinds of directed graphs obtained from public domain sparse matrix collections, showing promising spectral sparsification and partitioning results for general directed graphs.

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

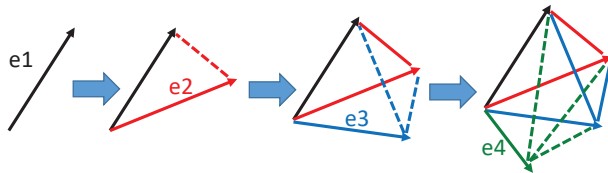

Figure 3: Edge coupling during directed Laplacian symmetrization.

## 7 APPENDIX

### 7.1 PROOF OF THEOREM 3.2

*Proof.* Each element $(i, j)$ in $\mathbf{L_{G_u}}$ can be written as follows:

$$L_{G_u}(i, j) = \begin{cases} D_G{}^2(i, i) + \sum_k A_G{}^2(k, i) & i = j \\ \sum_k \left( -A_G(k, i)A_G(k, j) + A_G(k, i)D_G(k, j) + D_G(k, i)A_G(k, j) \right) & i \neq j. \end{cases} \quad (20)$$

It can be shown that the following is always true:

$$L_{G_u}(i, i) + \sum_{j, j \neq i} L_{G_u}(i, j) = \sum_k L_G(i, k)L_G(i, k) + \sum_{j, j \neq i} \sum_k L_G(j, k)L_G(i, k)$$

$$= \sum_k L_G(i, k) \left( L_G(i, k) + \sum_{j, j \neq i} L_G(j, k) \right) = 0, \quad (21)$$

which indicates the all-one vector is the null space of $\mathbf{L_{G_u}}$. For directed graphs, it can be shown that if a node has more than one outgoing edge, in the worst case the neighboring nodes pointed by such outgoing edges will form a clique possibly with negative edge weights in the corresponding undirected graph after symmetrization.

As an example shown in Figure 3, when edge $e2$ is added into the initial graph $G$ that includes a single edge $e1$, an extra edge (shown in red dashed line) coupling with $e1$ will be created in the resultant undirected graph $G_u$; similarly, when an edge $e3$ is further added, two extra edges coupling with $e1$ and $e2$ will be created in $G_u$. When the last edge $e4$ is added, It forms a clique. Also, it can be shown that $G_u$ will contain negative edge weights under the following condition:

$$\sum_k (A_G(k, i)D_G(k, j) + D_G(i, k)A_G(j, k)) > \sum_k A_G(k, i)A_G(k, j). \quad (22)$$

In some cases, there may exist no clique even though all outgoing edges according to one node are added into subgraph only because the weights of these edges satisfy

$$\sum_k (A_G(k, i)D_G(k, j) + D_G(i, k)A_G(j, k)) = \sum_k A_G(k, i)A_G(k, j). \quad (23)$$

$\square$

### 7.2 EXISTENCE OF LINEAR-SIZED SPECTRAL SPARSIFIER FOR DIRECTED GRAPHS

Existing spectral sparsification methods for undirected graphs (Spielman & Teng, 2011; Batson et al., 2012; Feng, 2016) can not handle undirected graphs with negative-weighted edges. So we have to first prove the existence of spectral sparsifier for undirected graph with negative-weighted edges, or the existence of linear-sized spectral sparsifier for directed graphs .

$$B = \begin{bmatrix} 1 & 0 & -1 \\ 0 & -1 & 1 \\ 0 & 1 & -1 \\ -1 & 1 & 0 \end{bmatrix} \quad C = \begin{bmatrix} 1 & 0 & -1 \\ 0 & -1 & 1 \\ 0 & 1 & -1 \\ -1 & 1 & 0 \end{bmatrix}$$

$$W = \begin{bmatrix} 1 & 0 & 0 & 0 \\ 0 & 1 & 0 & 0 \\ 0 & 0 & 1 & 0 \\ 0 & 0 & 0 & 1 \end{bmatrix} \quad L_G = B^\top W C = \begin{bmatrix} 1 & -1 & 0 \\ 0 & 2 & -1 \\ -1 & -1 & 1 \end{bmatrix}$$

Figure 4: Forming a directed Laplacian with **B** and **C** matrices.

**Theorem 7.1.** *For a given directed graph $G$ and its undirected graph $G_u = (V, E_{G_u}, w_{G_u})$ obtained via Laplacian symmetrization, there exists a $(1 + \epsilon)$-spectral sparsifier $S$ with $O(n/\epsilon^2)$ edges such that its undirected graph $S_u = (V, E_{S_u}, w_{S_u})$ after symmetrization satisfies the following condition for any $\mathbf{x} \in \mathbb{R}^{\mathbf{n}}$:*

$$(1 - \epsilon)\mathbf{x}^\top \mathbf{L_{G_u}} \mathbf{x} \le \mathbf{x}^\top \mathbf{L_{S_u}} \mathbf{x} \le (1 + \epsilon)\mathbf{x}^\top \mathbf{L_{G_u}} \mathbf{x}. \tag{24}$$

Before proving **Theorem 7.1** for symmetrized undirected graph $G_u$ with negative edge weight, we need to introduce the following lemma 7.2 (Batson et al., 2012).

**Lemma 7.2.** *Let $d > 0$, and $\mathbf{u_1}, \mathbf{u_2}, ..., \mathbf{u_m}$ denote a set of vectors in $\mathbb{R}^n$ that allow expressing the identity decomposition as:*

$$\sum_{\mathbf{1 \le i \le m}} \mathbf{u_i u_i^\top} = \mathbf{id_{\mathbb{R}^n}}, \tag{25}$$

where $\mathbf{id_{\mathbb{R}^n}} \in \mathbb{R}^{n \times n}$ denotes the identity matrix. Then there exists a series of non-negative coefficients $\{t_i\}_{i=1}^m$ with $|i : t_i \ne 0| \le dn$ so that

$$\mathbf{x}^\top \mathbf{id_{\mathbb{R}^n}} \mathbf{x} \le \sum_i \mathbf{t_i x^\top u_i u_i^\top x} \le (1 + \epsilon)\mathbf{x}^\top \mathbf{id_{\mathbb{R}^n}} \mathbf{x}. \quad \forall \mathbf{x} \in \mathbb{R}^n \tag{26}$$

*Proof.* Lemma (7.2) proves the existence of the sparsifier for an undirected graph with positive weight edges. We have to extend the above lemma to prove the existence of the sparsifier for a symmetrized undirected graph with negative weight edges. The key of our approach is to construct a set of vectors $\mathbf{u_1}, \cdots, \mathbf{u_m}$ in $\mathcal{R}^n$ such that $\mathbf{u_i}$ can be expressed as an identity decomposition (25).

To construct $\mathbf{u_i}$, the Laplacian of an undirected graph can be written as $\mathbf{L_G} = \mathbf{B}^\top \mathbf{W B}$ (Spielman & Srivastava, 2011), where $\mathbf{B}_{m \times n}$ is the signed edge-vertex incidence matrix:

$$B(i, v) = \begin{cases} 1 & \text{if } v \text{ is i-th edge's head} \\ -1 & \text{if } v \text{ is i-th edge's tail} \\ 0 & \text{otherwise .} \end{cases} \tag{27}$$

$\mathbf{W_{m \times m}}$ is the diagonal matrix with $W(i, i) = w_i$. The Laplacian matrix of a directed graph can be written as $\mathbf{L_G} = \mathbf{B}^\top \mathbf{W C}$, where $\mathbf{C}_{m \times n}$ is a injection matrix defined as:

$$C(i, v) = \begin{cases} 1 & \text{if } v \text{ is i-th edge's head} \\ 0 & \text{if } v \text{ is i-th edge's tail} \\ 0 & \text{otherwise.} \end{cases} \tag{28}$$

Figure 4 shows an example for constructing a directed Laplacian matrix based on $B$ and $C$ matrices.

In the following, we show how to construct the vectors $\mathbf{u_i}$. The undirected Laplacian after symmetrization can be written as $\mathbf{L_{G_u}} = \mathbf{B}^\top \mathbf{W_o} \mathbf{B}$ with $\mathbf{W_o} = \mathbf{W}\mathbf{C}\mathbf{C}^\top \mathbf{W}$. Since $\mathbf{L_{G_u}}$ and its Pseudoinverse $\mathbf{L_{G_u}^+}$ can be written as

$$\mathbf{L_{G_u}} = \sum_{j=1}^{m} \lambda_j' \mathbf{u_j'}\mathbf{u_j'}^\top, \quad \mathbf{L_{G_u}^+} = \sum_{j=1}^{m} \frac{1}{\lambda_j'} \mathbf{u_j'}\mathbf{u_j'}^\top, \tag{29}$$

it can be shown that $\mathbf{L_{G_u}}\mathbf{L_{G_u}^+} = \sum_{j=1}^{m} \mathbf{u_j'}\mathbf{u_j'}^\top = \mathbf{id_{L_{G_u}}}$, where $\mathbf{id_{L_{G_u}}}$ is the identity on $\mathbf{im}(\mathbf{L_{G_u}}) = \mathbf{ker}(\mathbf{L_{G_u}})^\top$. Consequently, $\mathbf{U_{n \times m}}$ matrix with $\mathbf{u_i}$ for $i = 1, ..., m$ as its column vectors can be constructed as

$$\mathbf{U_{n \times m}} = [\mathbf{u_1}, ..., \mathbf{u_m}] = \mathbf{L_{G_u}^{+/2}}\mathbf{B}^\top \mathbf{W_o}^{1/2}. \tag{30}$$

It can be shown that $\mathbf{U_{n \times m}}$ will satisfy the following equation:

$$\mathbf{U_{n \times m}}\mathbf{U_{n \times m}}^\top = \sum_{i} \mathbf{u_i}\mathbf{u_i}^\top = \mathbf{L_{G_u}}^{+/2}\mathbf{B}^\top \mathbf{W_o}\mathbf{B}\mathbf{L_{G_u}}^{+T/2}$$
$$= \mathbf{L_{G_u}}^{+/2}\mathbf{L_{G_u}}\mathbf{L_{G_u}}^{+T/2} = \mathbf{id_{L_{G_u}}} \tag{31}$$

According to Lemma 7.2, we can always construct a diagonal matrix $\mathbf{T} \in \mathbb{R}^{\mathbf{m \times m}}$ with $t_i$ as its $i$-th diagonal element. Then there will be at most $O(n/\epsilon^2)$ positive diagonal elements in $\mathbf{T}$, which allows constructing $\mathbf{L_{S_u}} = \mathbf{B}^\top \mathbf{W_o}^{1/2}\mathbf{T}\mathbf{W_o}^{1/2}\mathbf{B}$ that corresponds to the directed subgraph $S$ for achieving $(1 + \epsilon)$-spectral approximation of $G$ as required by (24). It can be shown that each $\mathbf{u_i}$ with a nonzero $t_i$ coefficient corresponds to the outgoing edges pointed by the same node. Consequently, for directed graphs with bounded degrees, there will be $O(n/\epsilon^2)$ total number of directed edges in the $(1 + \epsilon)$-spectral sparsifier $S$. □

## 7.3 ALGORITHM FLOW

---

**Algorithm 2** Edge_Similarities_Checking
___

**Input:** $E_{\text{list}}$, $\mathbf{L_G}$, $\mathbf{L_S}$, $d_{\text{out}}$, $\varrho$

1: Perform t-step power iterations with $r = O(\log n)$ initial random vectors $\mathbf{h_0^{(1)}}, ..., \mathbf{h_0^{(r)}}$ to compute $r$ approximate dominant generalized eigenvectors $\mathbf{h_t^{(1)}}, ..., \mathbf{h_t^{(r)}}$;
2: Compute a $r$-dimensional embedding vector $\mathbf{T}_{p_i, q_i} \in \mathbb{R}^r$ for $\forall (p_i, q_i) \in E_{\text{list}}$;
3: let $E_{\text{addlist}} = [(p_1, q_1)]$;
4: **for** i=2:$|E_{\text{list}}|$ **do**
5:     Calculate the spectral similarity score $\beta_{i,j}$ between $(p_i, q_i)$ and every edge $(p_j, q_j)$ in $E_{\text{addlist}}$;
6:     **if** $1 - \beta_{i,j} < \varrho$, for $\forall (p_j, q_j) \in E_{\text{addlist}}$ **then**
7:         $E_{\text{addlist}} = [E_{\text{addlist}}; (p_i, q_i)]$;
8:     **end if**
9: **end for**
10: Return $E_{\text{addlist}}$ ;

---

## 7.4 RESULTS OF DIRECTED GRAPH SPARSIFICATION

Figure 5 shows the spectral sensitivities of all the off-subgraph edges ($e2$ to $e19$ represented with blue color) in both directed and undirected graphs calculated using MATLAB's "eigs" function and the proposed method based on (17) using the LAMG solver, respectively. Meanwhile, the spectral sensitivities of all the off-subgraph edges ($e2$ to $e19$) with respect to the dominant eigenvalues ($\lambda_{max}$ or $\lambda_1$) in both directed and undirected graphs are plotted. We observe that spectral sensitivities for directed and undirected graphs are drastically different from each other. The reason is that the spectral sensitivities for off-subgraph edges in the directed graph depend on the edge directions. It is also observed that the approximate spectral sensitivities calculated by the proposed $t$-step power iterations with the LAMG solver match the true solution very well for both directed and undirected graphs.

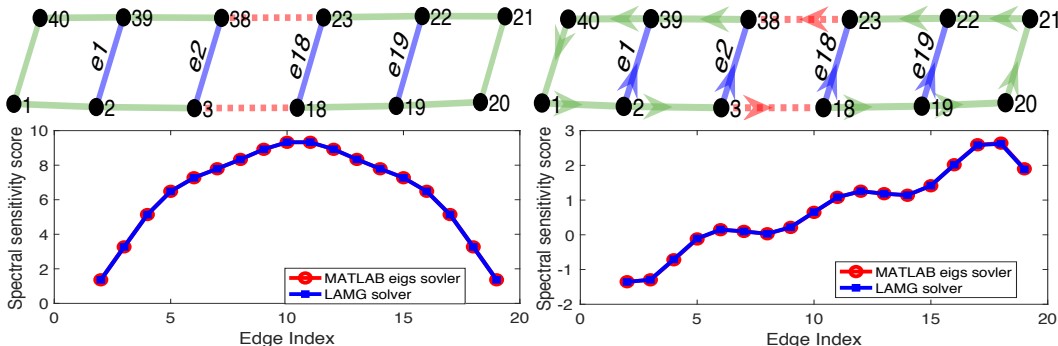

Figure 5: The spectral sensitivity scores of off-subgraph edges ($e2$ to $e19$ in blue) for the undirected (left) and directed graph (right).

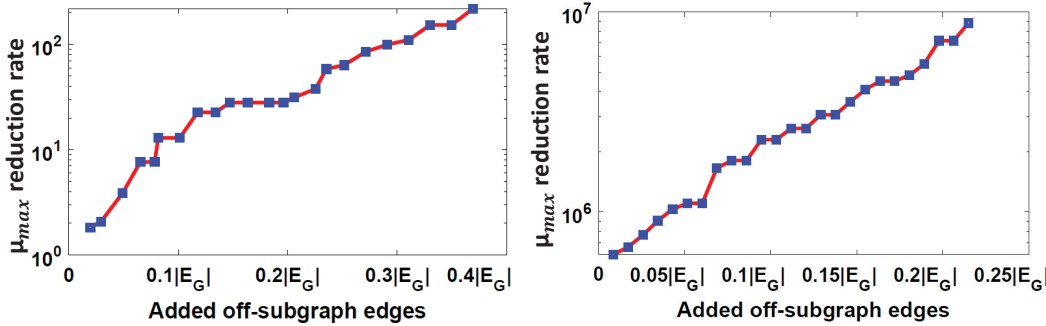

Figure 6: Largest generalized eigenvalue reduction rates for "gre_115" (left) and "pesa" (right).

We plot the detailed reduction rates of the largest generalized eigenvalue when adding different number of off-subgraph edges to the sparsifiers of graph "gre_115" and "peta" in Figure 6. It shows that the largest generalized eigenvalue can be effectively reduced if sufficient off-subgraph edges are included into the sparsifier.

## 7.5 APPLICATIONS IN DEVELOPING DIRECTED LAPLACIAN SOLVER

Consider the solution of the following linear systems of equations:

$$\mathbf{L}\mathbf{x} = \mathbf{b}. \tag{32}$$

Recent research has been focused on more efficiently solving the above problem when $\mathbf{L}$ is a Laplacian matrix of an undirected graph (Kelner et al., 2014; Koutis et al., 2010). In this work, we will mainly focus on solving nonsymmetric Laplacian matrices that correspond to directed graphs.

### 7.5.1 DIRECT METHOD FOR DIRECTED LAPLACIAN SOLVER

**Lemma 7.3.** *When solving (32), the right preconditioning system is applied, leading to the following alternative linear system of equations:*

$$\mathbf{L}_{\mathbf{G}_\mathbf{u}}\mathbf{y} = \mathbf{b}, \tag{33}$$

*where vector $\mathbf{b}$ will lie in the left singular vector space. When the solution of (33) is obtained, the solution of (32) is given by $\mathbf{L}_{\mathbf{G}}^{\top}\mathbf{y} = \mathbf{x}$.*

It is obvious that solving the above equation is equivalent to solving the problem of $\mathbf{L}_{\mathbf{G}}\mathbf{L}_{\mathbf{G}}^{\top}\mathbf{L}_{\mathbf{G}}^{+\top}\mathbf{x} = \mathbf{b}$. In addition, $\mathbf{L}_{\mathbf{G}_\mathbf{u}}$ is a Laplacian matrix of an undirected graph that can be much denser than $\mathbf{L}_{\mathbf{G}}$. Therefore, we propose to solve the linear system of $\mathbf{L}_{\mathbf{S}_\mathbf{u}}\tilde{\mathbf{y}} = \mathbf{b}$ instead to effectively approximate (33) since $G_{S_u}$ is sparser than $G_{G_u}$ and more efficient to solve in practice.

We analyze the solution errors based on the generalized eignvalue problem of $\mathbf{L}_{\mathbf{G}_\mathbf{U}}$ and $\mathbf{L}_{\mathbf{S}_\mathbf{U}}$. We have $\mathbf{V}\mathbf{L}_{\mathbf{G}_\mathbf{U}}\mathbf{V}^{\top} = \lambda$ and $\mathbf{V}\mathbf{L}_{\mathbf{S}_\mathbf{U}}\mathbf{V}^{\top} = \mathbf{I}$, where $\mathbf{V} = [\mathbf{v_1}, \mathbf{v_2}, ..\mathbf{v_n}]$, $\lambda$ is the diagonal matrix with

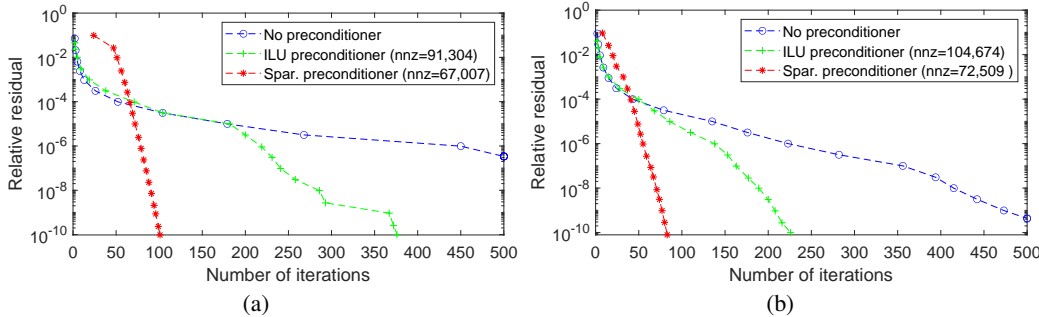

Figure 7: GMRES convergence results for graphs of (a) pesa and (b) big

its generalized eigenvalues $\lambda_i \geq 1$ on its diagonal. Since the errors can be calculated from the following procedure:

$$\mathbf{L_{G_U}}\mathbf{y} - \mathbf{L_{S_U}}\tilde{\mathbf{y}} = \mathbf{L_{G_U}}(\mathbf{y} - \tilde{\mathbf{y}}) + (\mathbf{L_{G_U}} - \mathbf{L_{S_U}})\tilde{\mathbf{y}} = \mathbf{0}, \tag{34}$$

we can write the error term as follows:

$$(\tilde{\mathbf{y}} - \mathbf{y}) = \mathbf{L_{G_U}^+}(\mathbf{L_{G_U}} - \mathbf{L_{S_U}})\tilde{\mathbf{y}}. \tag{35}$$

Since $\tilde{\mathbf{y}} = \sum_i \mathbf{a_i}\mathbf{v_i}$, the error can be further expressed as

$$(\tilde{\mathbf{y}} - \mathbf{y}) = \sum_i \mathbf{a_i}(1 - \frac{1}{\lambda_i})\mathbf{v_i}. \tag{36}$$

Therefore, the error term (36) can be generally considered as a combination of high-frequency errors (generalized eigenvectors with respect to high generalized eigenvalues) and low-frequency errors (generalized eigenvectors with respect to low generalized eigenvalues). After applying GS relaxations, the high-frequency error terms can be efficiently removed (smoothed), while the low-frequency errors tend to become zero if the generalized eigenvalues approach 1 considering $(1 - \frac{1}{\lambda_i})$ tends to be approaching zero. As a result, the error can be effectively eliminated using the above solution smoothing procedure.

In summary, in the proposed directed Laplacian solver, the following steps are needed:

(a) We will first extract a spectral sparsifier $\mathbf{L_S}$ of a given (un)directed graph $\mathbf{L_G}$. Then, it is possible to compute an approximate solution by exploiting its spectral sparsifier $\mathbf{L_{S_u}} = \mathbf{L_S}\mathbf{L_S^\top}$ via solving $\tilde{\mathbf{y}} = \mathbf{L_{S_u}^+}\mathbf{b}$ instead.

(b) Then we improve the approximate solution $\tilde{\mathbf{y}}$ by getting rid of the high-frequency errors via applying a few steps of GS iterations (Briggs, 1987).

(C) The final solution is obtained from $\mathbf{x} = \mathbf{L_G^\top}\tilde{\mathbf{y}}$.

Table 5: Relative errors between exact and approximate solutions of $\mathbf{L_G}\mathbf{x} = \mathbf{b}$ w/ or w/o smoothing

| Test Cases | gre_115 | gre_185 | cell1 | pesa | big | gre_1107 | wordnet3 |
|---|---|---|---|---|---|---|---|
| w/o smooth. | 0.41 | 0.42 | 0.44 | 2.1E-4 | 4.3E-3 | 0.6 | 0.72 |
| w/ smooth. | 0.04 | 0.12 | 0.07 | 8.0E-9 | 1.1E-4 | 0.10 | 0.07 |

Table 5 shows the results of the directed Laplacian solver on different directed graphs. It reports relative errors between the exact solution and the solution calculated by the proposed solver with and without smoothing. It shows that errors can be dramatically reduced after smoothing, and our proposed solver can well approximate the true solution of $\mathbf{L_G}\mathbf{x} = \mathbf{b}$

Figure 8: The correlation of PageRank between itself and its sparsifier for graph ibm_32 (left), mathworks_100 (middle) and gre_1107 (right) after smoothing

### 7.5.2 ITERATIVE METHOD FOR DIRECTED LAPLACIAN SOLVER

Figure 7 shows the relative residual plot (versus GMRES iteration number) when no preconditioner is applied, Incomplete LU factorization (ILU) as the preconditioner is applied, and the directed sparsifier Laplacian $\mathbf{L_S}$ as the preconditioner is applied for graph pesa and big. We can conclude that GMRES with directed sparsifiers as preconditioners has faster convergence rate than the other two methods. It is also observed that the number of nonzeros (nnz) in the preconditioner matrix created by the directed sparsifier is the lowest.

### 7.6 APPLICATIONS IN COMPUTING (PERSONALIZED) PAGERANK VECTORS

The idea of PageRank is to give a measurement of the importance for each web page. For example, PageRank algorithm aims to find the most popular web pages, while the personalized PageRank algorithm aims to find the pages that users will most likely to visit. To state it mathematically, the PageRank vector $\mathbf{p}$ satisfies the following equation:

$$\mathbf{p} = \mathbf{A_G^\top D_G^{-1} p}, \tag{37}$$

where $\mathbf{p}$ is also the eigenvector of $\mathbf{A_G^\top D_G^{-1}}$ that corresponds to the eigenvalue equal to $\mathbf{1}$. Meanwhile, $\mathbf{p}$ represents the stable distribution of random walks on graph $G$. However, $\mathbf{D_G^{-1}}$ can not be defined if there exists nodes that have no outgoing edges. To deal with such situation, a self-loop with a small edge weight can be added for each node.

The stable distributions of (un)directed graphs may not be unique. For example, the undirected graphs that have multiple strongly-connected components, or the directed graphs that have nodes without any outgoing edges, may have non-unique distributions. In addition, it may take very long time for a random walk to converge to a stable distribution on a given (un)directed graph.

To avoid such situation in PageRank, a jumping factor $\alpha$ that describes the possibility at $\alpha$ to jump to a uniform vector can be added, which is shown as follows:

$$\mathbf{p} = (1 - \alpha)\mathbf{A_G^\top D_G^{-1} p} + \frac{\alpha}{n}\mathbf{1}, \tag{38}$$

$$\mathbf{p} = \frac{\alpha}{n}(\mathbf{I} - (1 - \alpha)\mathbf{A_G}^\top \mathbf{D_G^{-1}})^{-1}\mathbf{1}, \tag{39}$$

where $\alpha \in [0, 1]$ is a jumping constant. After applying Taylor expansions, we can obtain that

$$\mathbf{p} = \frac{\alpha}{n}\sum_i ((1 - \alpha)\mathbf{A_G^\top D_G^{-1}})^\mathbf{i}. \tag{40}$$

By setting the proper value of $\alpha$ (e.g., $\alpha = 0.15$), the term $(1 - \alpha)^i$ will be quickly reduced with increasing $i$. Instead of starting with a uniform vector $\frac{\alpha}{n}\mathbf{1}$, a nonuniform personalization vector $\mathbf{pr}$ can be applied:

$$\mathbf{p} = (1 - \alpha)\mathbf{A_G^\top D_G^{-1} p} + \alpha\mathbf{pr}. \tag{41}$$

Figure 8 shows the application of the proposed directed graph sparsification for computing PageRank vectors, where the correlation of PageRank results using the original graphs (x-axis) and sparsifiers (y-axis) are plotted for graph ibm_32 (left), mathworks_100 (middle) and gre_1107 (right). Note

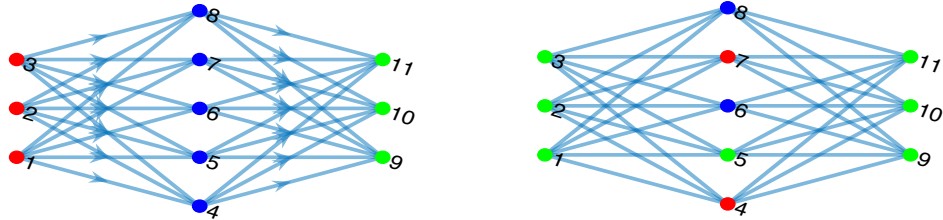

Figure 9: The correlation of personalized PageRank between itself and its sparsifier for graph ibm_32 (left), mathworks_100 (middle) and gre_1107 (right) after smoothing

Figure 10: Spectral partitioning of directed (left) and undirected graphs (right). The nodes within the same cluster are assigned the same color.

that a few steps of Gauss-Seidel smoothing have been applied to remove the high-frequency errors to obtain the smoothed PageRank vectors when using the sparsified graphs. We observe that the PageRank vectors obtained from sparsifiers can well approximate the results computed with the original graphs.

Similar to the results of PageRank in Figure 8, Figure 9 shows the application of the proposed directed graph sparsification on the personalized PageRank, where the correlations of personalized PageRank results using the original graphs (x-axis) and sparsifiers (y-axis) are plotted for graph ibm_32 (left), mathworks_100 (middle) and gre_1107 (right). Gauss-Seidel smoothing are also applied when using the sparsified graphs. We can observe that personalized PageRank vectors from sparsifiers match very well with the ones generated from original graphs, which demonstrates the effectiveness of the sparsifiers on the Personalized PageRank application.

## 7.7 Applications in directed graph partitioning

It has been shown that partitioning and clustering of directed graphs can play very significant roles in a variety of applications related to machine learning (Malliaros & Vazirgiannis, 2013), data mining and circuit synthesis and optimization (Micheli, 1994), etc. However, the efficiency of existing methods for partitioning directed graphs strongly depends on the complexity of the underlying graphs (Malliaros & Vazirgiannis, 2013).

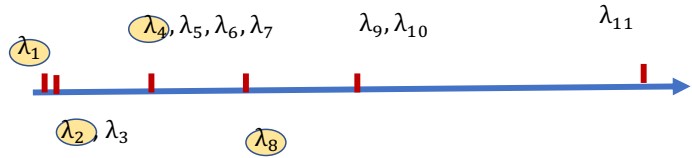

Figure 11: Eigenvalues distribution of $L_{G_U}$ for the directed graph in Figure 10

In this work, we propose a spectral method for directed graph partitioning problems. For an undirected graph, the eigenvectors corresponding to the first few smallest eigenvalues can be utilized for the spectral partitioning purpose (Spielman & Teng, 1996). For a directed graph $G$ on the other hand, the left singular vectors of Laplacian $\mathbf{L_G}$ will be required for directed graph partitioning. The

eigen-decomposition of its symmetrization $\mathbf{L_{G_U}}$ can be wirtten as

$$\mathbf{L_{G_U}} = \sum_i \lambda_i \mathbf{v_i} \mathbf{v_i}^\top,$$ (42)

where $0 = \lambda_1 \leq ... \lambda_k$ and $\mathbf{v_1}, ..., \mathbf{v_k}$, with $k \leq n$ denote the Laplacian eigenvalues and eigenvectors, respectively. There may not be $n$ eigenvalues when there are some nodes without any outgoing edges. In addition, the spectral properties of $\mathbf{L_{G_U}}$ are more complicated since the eigenvalues always have multiplicity (either algebraic or geometric multiplicities). For example, the eigenvalues according to the symmetrization of the directed graph in Figure 10 have a a few multiplicities: $\lambda_2 = \lambda_3$, $\lambda_4 = \lambda_5 = \lambda_6 = \lambda_7$, $\lambda_9 = \lambda_{10}$, as shown in Figure 11.

Therefore, we propose to exploit the eigenvectors (left singular vectors of directed Laplacian) corresponding to the first few different eigenvalues (singular values of directed Laplacian) for directed graph partitioning. For example, the partitioning result of the directed graph in Figure 10 will depend on the eigenvectors of $\mathbf{v_1}, \mathbf{v_2}, \mathbf{v_4}, \mathbf{v_8}$ that correspond to eigenvalues of $\lambda_1, \lambda_2, \lambda_4, \lambda_8$. As shown in Figure 10, the spectral partitioning results can be quite different between the directed and undirected graph with the same set of nodes and edges.

In general, it is possible to first extract a spectrally-similar directed graph before any of the prior partitioning algorithms are applied. Since the proposed spectral sparsification algorithm can well preserve the structural (global) properties of the original graphs, the partitioning results obtained from the sparsified graphs will be very similar to the original ones.

Figure 12 shows the spectral graph partitioning results on the symmetrized graph $G_u$ of original directed graph and its symmetrized sparsifier $S_u$. As observed, very similar partitioning results have been obtained, indicating well preserved spectral properties within the spectrally-sparsified directed graph.

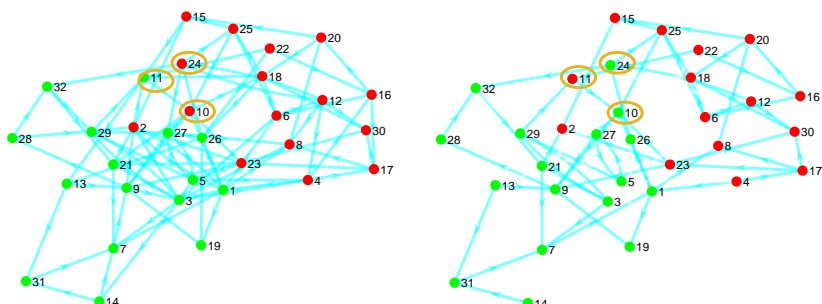

Figure 12: The partitioning results between $G_u$ (left) and its sparsifier $S_u$ (right) for the 'ibm32.mtx' graph.

