# OpenReview forum: "A Unified Spectral Sparsification Framework for Directed Graphs"
_ICLR.cc/2021/Conference — Reject_

### Official Review · AnonReviewer1 · 2020-10-24

**Rating:** 3
**Confidence:** 3

**Review:**

Summary
========
The paper studies a certain notion of spectral sparsification of directed graphs. It claims the existence of nearly linear sized sparsifiers under this notion, and suggests empirical methods to produce such sparsifiers in nearly linear time.

Comments
=========

Section 4: I am not sure what is the downstream justification for the notion of spectral approximation studied here (relative condition number) in the context of directed graphs, and it would help if the authors could elaborate on this. For example, the notion defined in Cohen et al. (2017) enables fast approximate solution of linear equations, which is a primitive in various algorithmic tasks on directed graphs. The notion in this paper does not rigorously lead to such consequences (or at least none are presented), which leaves the question why one should be interested in an existence result like Theorem 4.3. (The appendix contains some experimental results for linear system solving, but no rigorous claims.)

About the proof Theorem 4.3: Could you please clarify how does the approximating matrix $L_{S_u} = B^T W_o^{1/2} T W_o^{1/2} B$ (defined in the end of section 8.2) give rise to a directed subgraph? How does one construct $S$ (or in other words, why can $L_{S_u}$ be written as $L_SL_S^T$ for $L_S$ of the form in eq (18))

Section 5: I am generally not able to follow the derivations in section 5.2, and it may be needed to be written more clearly. Specific comments/questions:
1. How come $\lambda_n>0$ if the matrix $L_{S_u}^+L_{G_u}$  has non-full rank?
2. You define v_i once as the eigenvectors of $L_{S_u}^+L_{G_u}$ in the first sentence sentence as then as scaled eigenvalues of $L_{S_u}$ between eq (5) and (6), which one is it? Or are they supposed to be denoted differently?
3. Could you elaborate on how you arrive at eq (5)?

Section 6: The experiments in the main text concentrate on measuring the relative condition number of the sparsifier w.r.t the original graph as per the sparsification notion studied in this paper, but as written above it is not clear what do we actually get out of the sparsifier. In general I have doubts about fit to the venue; while ICLR scope is broad and inclusive and spectral sparsification has certain potential connections to ML, this paper does not highlight any of them, and it is not entirely clear what it is attempting to achieve.

Conclusion:
=========
Pros: Spectral sparsification of directed graphs is a relatively new field, so far with initial results that invite further research and improvements. The paper implements an algorithm which seems to improve over baselines under certain metrics.
Cons: It is not clear what ML task the paper is trying to solve or improve upon, and what improvement is achieved. Viewed as a theoretical submission, the analysis is unclear to me in many parts, and at present I am unable to confirm its soundness. The algorithm implemented in the experimental section seems rather detached from the preceding theoretical definitions, whose purpose remains somewhat unclear.

Post-discussion update: I thank the authors for their participation in the discussion. Unfortunately I find it mostly has not cleared the numerous question marks I have regarding the paper. I recommend putting more effort into clarifying the mathematical derivations and into positioning the paper correctly w.r.t. prior work on the topic.

---

> ### Author Response · Authors · 2020-11-18
> **More clear explanations have been provided in the latest draft**
>
> "Section 4: I am not sure what is the downstream justification for the notion of spectral approximation studied here (relative condition number) in the context of directed graphs, and it would help if the authors could elaborate on this. For example, the notion defined in Cohen et al. (2017) enables fast approximate solution of linear equations, which is a primitive in various algorithmic tasks on directed graphs. The notion in this paper does not rigorously lead to such consequences (or at least none are presented), which leaves the question why one should be interested in an existence result like Theorem 4.3. (The appendix contains some experimental results for linear system solving, but no rigorous claims.)
> About the proof Theorem 4.3: Could you please clarify how does the approximating matrix  (defined in the end of section 8.2) give rise to a directed subgraph? How does one construct  (or in other words, why can  be written as  for  of the form in eq (18))"
>
> ----Our response: Thanks a lot for the feedback. We have included more information in the latest paper to show more detailed procedures for the construction of directed sparsifiers. A more detailed proof of our theorem is also provided.
>
> "Section 5: I am generally not able to follow the derivations in section 5.2, and it may be needed to be written more clearly. Specific comments/questions:
> How come  if the matrix   has non-full rank?"
>
> ----Our response: The directed or undirected Laplacians are all non-full rank matrices. When solving the generalized eigenvalue problems in section 5.3, we add a very small value into the diagonal of L_{S_u} to make L_{S_u} a full-rank matrix, which is equivalent to introducing a rather weak self-loop to each node. Such a modification will not impact the global structure of the graph, and thus allows the proposed sparsification to be effectively performed.
>
> "You define v_i once as the eigenvectors of in the first sentence as then as scaled eigenvalues of between eq (5) and (6), which one is it? Or are they supposed to be denoted differently?
> eq (5) is derived from eq(4), and eq (6) represents that generalized eigenvectors  v_i are L_{s_u} orthogonal to each other, which is one of the properties for the generalized eigenvalue problems.  So v_i are not scaled during the whole procedure.
> Could you elaborate on how you arrive at eq (5)?"
>
> ----Our response: The detailed derivation of eq (5) has been updated in the latest paper. Hopefully, this will help address your concerns.
>
> "Section 6: The experiments in the main text concentrate on measuring the relative condition number of the sparsifier w.r.t the original graph as per the sparsification notion studied in this paper, but as written above it is not clear what do we actually get out of the sparsifier. In general I have doubts about fit to the venue; while ICLR scope is broad and inclusive and spectral sparsification has certain potential connections to ML, this paper does not highlight any of them, and it is not entirely clear what it is attempting to achieve."
>
> ----Our response: This work introduces the first unified framework for general spectral graph sparsification tasks. We show comprehensive applications of spectral sparsifiers of directed graphs: for example directed Laplacian solver can be used for computing the stationary distributions of Markov chains; directed graph partitioning and clustering are very important applications related to machine learning and data mining; PageRank and personalized PageRank play important roles in modeling and analysis of large-scale real-world networks. More detailed discussions about the applications of spectral sparsification of directed graphs can be found in the following papers:
>
> Michael B Cohen, et al.   Almost-linear-time algorithms for Markov chains and new spectral primitives for directed graphs. InProceedings of the 49th Annual ACM SIGACT Symposium on Theory of computing, pp. 410–419. ACM, 2017.
>
> Michael B. Cohen, et al. Solving Directed Laplacian Systems in Nearly-Linear Time through Sparse LU Factorizations. Foundations of Computer Science (FOCS), 2018 59th Annual IEEE Symposium on, pp. 898–909. IEEE

---

> > ### Comment · AnonReviewer3 · 2020-11-22
> > **re: references in last comment**
> >
> > (from reviewer 3, responding here because the same response was posted there, so am hoping to do things in a more aggregated manner)
> >
> > It seems that both reviewer 1 and I are concerned about the relevance of directed graph/Laplacian solvers in ML. The additional citations provided are from theory venues, and only discuss using these solvers to compute related quantities such as hitting times, escape probabilities, and commute times. I'm not aware of any of these quantities being used in large scale applications.
> >
> > Also, I believ Pagerank is often used with fairly large reset probability, which in turn allow for significantly simpler algorithms.
> >
> > Are you aware of ML applications involving directed graphs with significantly higher mixing times? I would be open to change my review if pointers are given to practically relevant, highly ill-conditioned, directed Laplacians.

---

> > > ### Author Response · Authors · 2020-11-23
> > > **Some recent works that exploited graph sparsification techniques**
> > >
> > > Thanks for your suggestions and comments. Please find the following related works published in recent ICML conferences:
> > > (1) Improved large-scale graph learning through ridge spectral sparsification, ICML’2018
> > > (2) Improved Dynamic Graph Learning through Fault-Tolerant Sparsification, ICML’2019
> > > (3) Robust Graph Representation Learning via Neural Sparsification, ICML’ 2020
> > > The following recent works show that spectral graph sparsification can substantially improve the performance of graph neural networks (GNNs).
> > > (4) Fast Graph Attention Networks Using Effective Resistance Based Graph Sparsification. arXiv preprint arXiv:2006.08796 (2020).
> > > (5) SGCN: A Graph Sparsifier Based on Graph Convolutional Networks. In Pacific-Asia Conference on Knowledge Discovery and Data Mining (2020).
> > >
> > > Note that recent research works are mostly based on (spectral) sparsification of undirected graphs, while this work will potentially allow extending the prior graph-based ML applications to directed graphs.

---

> > ### Comment · AnonReviewer1 · 2020-11-23
> > **Response**
> >
> > I thank the authors for their response. I did not find in it answers to some of the primary questions I had raised --- apologies if I have missed them or misunderstood --- so let me highlight them again, hoping that the authors could kindly clarify them once more.
> >
> > 1. What are the formal implications of your notion of sparsification? You write correctly that Cohen et al. point out several formal application of directed graph sparsification, however, their definition of sparsifiers is different than yours. They prove that their definition formally leads to fast approximate solutions to graphic linear systems, which is a subroutine in numerous applications. Does your definition also give rise to this kind of formal result? Or is the connection to the experimental tasks only heuristic?
> >
> > 2. I still do not see how one extracts a directed subgraph from the matrix $L_{S_u}$ defined in the end of the proof of Theorem 7.2. It satisfies eq. (24) as a matrix, but why is it a sparsifier?

---

> > > ### Author Response · Authors · 2020-11-24
> > > **Our latest response**
> > >
> > > What are the formal implications of your notion of sparsification? You write correctly that Cohen et al. point out several formal application of directed graph sparsification, however, their definition of sparsifiers is different than yours. They prove that their definition formally leads to fast approximate solutions to graphic linear systems, which is a subroutine in numerous applications. Does your definition also give rise to this kind of formal result? Or is the connection to the experimental tasks only heuristic?
> > >
> > > ---Our response: as described on page 4 (Section 3.2) of our paper, for directed graphs our method will extract a subgraph $S$ such that the condition number of $L_S^+L_G$ will approach 1, or equivalently, both the max/min singular values of $L_S^+L_G$ will approach 1. This property also implies very similar spectral properties between the subgraph $S$ and the original graph $G$. A small condition number (close to 1) allows iterative methods, such as the GMRES algorithm, to converge very quickly, which leads to the development of fast Laplacian solvers for directed graphs. So, all our experimental results are clearly connected with our theory.
> > >
> > >
> > > I still do not see how one extracts a directed subgraph from the matrix
> > >  defined in the end of the proof of Theorem 7.2. It satisfies eq. (24) as a matrix, but why is it a sparsifier?
> > >
> > > ---Our response: as shown in Section 4.2 in the revised draft, we formulate the spectral sparsification task as a min-max optimization problem in (8). After converting the directed graphs ($S$ and $G$) into undirected ones ($S_u$ and $G_u$) using the proposed spectrum-preserving Laplacian symmetrization process, in our approach the following two steps will be performed iteratively such that the maximum mismatch (that is bounded by the largest eigenvalue $\lambda_1$) between graphs $S_u$ and $G_u$ will be minimized: (Step A) we first find out the boundary (or the node set) that has the greatest cut mismatch between  ${S_u}$ and ${G_u}$ based on spectral edge embedding with the dominant generalized eigenvector, as described in equations (5) to (7); (Step B) Next, we add a few edges (in $G$) to the subgraph (S) so that the largest cut mismatch (the largest eigenvalue $\lambda_1$) between $S_u$ and $G_u$ can be drastically reduced, as described in equation (8). By repeatedly applying the above two-step procedure, the largest eigenvalue ($\lambda_1$) will become very small, implying very small mismatches between the subgraph $S$ and the original graph $G$. To achieve the above, Section 4.3 is provided with more details.

---

### Official Review · AnonReviewer3 · 2020-10-28
**Contributions Unclear**

**Rating:** 5
**Confidence:** 4

**Review:**

This paper introduces new notions of producing small approximations of directed graphs. These notions are based on measuring importances of edges with respect to quadratic forms of symmetrizations of the Laplacian. The algorithm is implemented, and some experimental results for the effectiveness of the sparse approximation as preconditioners are shown.

I feel the contributions of the paper are more in the numerical analysis / linear systems solving setting. The new notion introduced appears fundamentally similar to the error-after-symmetrizatzion approaches taken in previous works. The experimental results don't involve end-to-end uses of directed Laplacians / directed random walks for learning tasks, but are more about approximations for solutions of linear systems / pagerank produced. Furthermore, I'm unfamiliar with uses of methods such as GMRES in learning tasks. So given such concerns, I'm unsure how relevant this paper is to ICLR.

---

> ### Author Response · Authors · 2020-11-18
> **Spectral sparsification is a fundamental tool that may benefit a wide range of applications**
>
> "I feel the contributions of the paper are more in the numerical analysis / linear systems solving setting. The new notion introduced appears fundamentally similar to the error-after-symmetrizatzion approaches taken in previous works. The experimental results don't involve end-to-end uses of directed Laplacians / directed random walks for learning tasks, but are more about approximations for solutions of linear systems / pagerank produced. Furthermore, I'm unfamiliar with uses of methods such as GMRES in learning tasks. So given such concerns, I'm unsure how relevant this paper is to ICLR."
>
> ----Our response: This work introduces the first unified framework for general spectral graph sparsification tasks. We show comprehensive applications of spectral sparsifiers of directed graphs: for example directed Laplacian solver can be used for computing the stationary distributions of Markov chains; directed graph partitioning and clustering are very important applications related to machine learning and data mining; PageRank and personalized PageRank play important roles in modeling and analysis of large-scale real-world networks. More detailed discussions about the applications of spectral sparsification of directed graphs can be found in the following papers:
>
> Michael B Cohen, et al.   Almost-linear-time algorithms for Markov chains and new spectral primitives for directed graphs. InProceedings of the 49th Annual ACM SIGACT Symposium on Theory of computing, pp. 410–419. ACM, 2017.
>
> Michael B. Cohen, et al. Solving Directed Laplacian Systems in Nearly-Linear Time through Sparse LU Factorizations. Foundations of Computer Science (FOCS), 2018 59th Annual IEEE Symposium on, pp. 898–909. IEEE

---

### Official Review · AnonReviewer2 · 2020-10-29
**Spectral Sparisifiers for Directed Graphs**

**Rating:** 4
**Confidence:** 2

**Review:**

Authors propose a novel method to approximate a given directed graph with a´nother one (the sparsifier) which has fewer edges. The proposed method seems promising. however, the paper needs significant improvement in clarity and contextualizing the results.

1. The main algorithmic result, Algorithm 1 should be placed in the main sectino of the paper and not an appendix.
2. The main numerfic results, e.g., applying the sparsifier to linear solvers should be put in the main section of the paper and not hidden in the appendix.
3. The comparision for the linear solver application should also inclucde the spcial case of undirected graphs. how does your sparsifier compare then with Spielmen et.al. mehtods. Also, for the general case of directed graphs which are not strongly connected, you should compare with existing sparsifiers for directed graphs but whose analysis requires them to be strongly connected. Maybe in practice those existing sparsifiers work fine.
4. Pls also evaluate the sparsifier on machine learning applications, e.g., federated and mulit-task learning over networks:

A. Jung and N. Tran, "Localized Linear Regression in Networked Data," in IEEE Signal Processing Letters, vol. 26, no. 7, pp. 1090-1094, July 2019, doi: 10.1109/LSP.2019.2918933.

minor issues:
- "..in which any node can be reached from any other node along with direction."
- "...will have the all-one vector as its null space,..."  --> the null space consists of more than one single vector

---

> ### Author Response · Authors · 2020-11-18
> **Substantial changes have been made based on your valuable suggestions**
>
> "The main algorithmic result, Algorithm 1 should be placed in the main section of the paper and not an appendix.
> The main numeric results, e.g., applying the sparsifier to linear solvers should be put in the main section of the paper and not hidden in the appendix."
>
> ----Our response: Thanks for the advice for re-organizing the materials of this paper. The core algorithm and most important numerical results have been moved to the main paper.
>
> "The comparison for the linear solver application should also include the special case of undirected graphs. How does your sparsifier compare then with Spielman et.al. methods? Also, for the general case of directed graphs which are not strongly connected, you should compare with existing sparsifiers for directed graphs but whose analysis requires them to be strongly connected. Maybe in practice, those existing sparsifiers work fine."
>
> ----Our response: Thanks for the suggestions. We have compared with another undirected graph sparsification tool GRASS, showing that ignoring edge directivities will lead to much worse results. We also tried to use Spielman's implementation that is available at https://github.com/danspielman/Laplacians.jl. However, the program requires a rather nontrivial tuning of the input parameters for sampling edges using effective resistance, which is also dataset dependent. As a result, for many occasions, it failed to produce a desirable sparsity in the output graph. We expect this tool if optimally tuned, would produce a similar result as GRASS which we have used in our current comparisons.
>
> "Pls also evaluate the sparsifier on machine learning applications, e.g., federated and multi-task learning over networks:
> A. Jung and N. Tran, "Localized Linear Regression in Networked Data," in IEEE Signal Processing Letters, vol. 26, no. 7, pp. 1090-1094, July 2019, DOI: 10.1109/LSP.2019.2918933."
>
> ----Our response: Thanks for the suggestions. The suggested work targets a quite different research task that aims to solve the network Lasso (nLasso) problem involving only undirected graphs. It is not very clear to us if the properties preserved by spectral sparsification can directly benefit such applications, and more importantly, result in a substantial improvement of the solution. We believe it will be valuable for us to spend more time studying the suggested problems in the future.

---

### Official Review · AnonReviewer4 · 2020-10-31
**Spectral Sparsification for Directed Graphs**

**Rating:** 7
**Confidence:** 4

**Review:**

This paper considers the problem of sparsifying directed graphs, a timely task of importance in many applications. The main contribution is the proposal of a unified approach for spectral sparsification of such directed graphs.

The proposed eigenvalue perturbation pipeline is interesting in its own right, and similarly that of ranking edges in terms of spectral importance from Section 5.2.  One of the main advantages of the approach is that it does not require the underlying directed graphs to be strongly connected and aperiodic, unlike recent work from the literature. Their proposed approach is fairly straightforward, but yet appears to be very efficient. The authors do a good job of placing their work in the relevant context, highlighting drawbacks of existing symmetrization techniques. The authors rightfully point out that naive symmetrization schemes from the literature may damage the structural properties of the graph. The proposed a Laplacian symmetrization approach which enjoys theoretical guarantees while remaining computationally scalable. The authors also mention other properties of interest that one could preserve during the sparsification process.

The authors also evaluate the embedding given by the proposed symmetrized Laplacian in the context of clustering directed graphs, where the majority of existing spectral methods tend to underperform. Clustering directed graphs is an important timely problem, less studied, and the present work makes also a nice contribution in that regard. The application to computing the (Personalized) PageRank is also nice (the sparsification appears to perform well in the PageRank computation), but perhaps less impactful since since this can be computed fast via a number of other methods.

The authors could further provide more intuition linking Theorem 4.3 on the existence of the sparsifier, and Section 5, in particular the pipeline in Section 5.2  The authors could also consider testing on synthetic data sets, also to assess the behaviour of the proposed method on graphs with increasing density. What about comparison with other approaches? The authors claim strong performance in their numerical results, but it would be good to have some baselines to compare against.

---

> ### Author Response · Authors · 2020-11-18
> **A better connection between spectral sparsification and the spectral perturbation framework**
>
> "The authors could further provide more intuition linking Theorem 4.3 on the existence of the sparsifier, and Section 5, in particular, the pipeline in Section 5.2 The authors could also consider testing on synthetic data sets, also to assess the behavior of the proposed method"
>
> ----Our response: Thanks a lot for the feedback. We have included more information (detailed proof for Theorem 7.1) about the existence of the directed graph sparsifier in Section 7.2 in the Appendix. We also provided a more clear discussion about using dominant generalized eigenvalues and eigenvectors for constructing spectral sparsifiers.

---

### Author Response · Authors · 2020-11-18
**The revised paper has addressed major concerns from the reviewers**

We would like to thank the reviewers for providing so constructive suggestions for this work. To make the paper more accessible to the general audience, we have made substantial changes to the previous draft. In this revised draft, we cast the general spectral (directed or undirected) graph sparsification problem into a Riemannian distance minimization problem. We show that as long as we can find a sparse subgraph such that its Laplacian has a small Riemannian Distance from the original Laplacian, the two graphs will have similar graph spectral properties. To this end, we introduce a min-max optimization formulation (8) for tackling the Riemannian distance minimization problems. Next, a spectral perturbation framework is introduced to allow efficiently solving the above min-max optimization problem. We also provide more detailed procedures for the proof of Theorem 7.1. Some minor changes are also made based on the reviewers' suggestions: we have moved some important contents/results to the main section of the paper.

---

### Decision · Program_Chairs · 2021-01-07
**Final Decision**

**Decision:**

Reject

**Comment:**

The paper proposes a fast, nearly-linear time, algorithm for finding a sparsifier for general directed and undirected graphs that approximately preserves the spectral properties of the original graph. The reviewers appreciated the main contribution of the paper, but they were concerned about the correctness and clarity of the paper, as well as the relevance of the contribution to machine learning. Following the discussion with the authors, the reviewers still felt that these concerns had not been fully addressed by the authors' responses and the subsequent revision of the paper. After taking these concerns into account as well as evaluating the paper relative to other ICLR submissions, I recommend reject.